# Influence of *Ziziphus lotus* (Rhamnaceae) Plants on the Spatial Distribution of Soil Bacterial Communities in Semi-Arid Ecosystems

**DOI:** 10.3390/microorganisms13081740

**Published:** 2025-07-25

**Authors:** Nabil Radouane, Zakaria Meliane, Khaoula Errafii, Khadija Ait Si Mhand, Salma Mouhib, Mohamed Hijri

**Affiliations:** 1African Genome Center, University Mohammed VI Polytechnic (UM6P), Lot 660, Hay Moulay Rachid, Ben Guerir 43150, Morocco; nabil.radouane-ext@um6p.ma (N.R.); zakaria.meliane@usherbrooke.ca (Z.M.); khaoula.errafii@um6p.ma (K.E.); khadija.aitsimhand@um6p.ma (K.A.S.M.); salma.mouhib@um6p.ma (S.M.); 2Institut de Recherche en Biologie Végétale, Département de Sciences Biologiques, Université de Montréal, 4101 Rue Sherbrooke Est, Montréal, QC H1X 2B2, Canada

**Keywords:** alpha-diversity, arid ecosystems, beta-diversity, microbial diversity, soil bacterial communities, spatial gradients, *Ziziphus lotus*

## Abstract

*Ziziphus lotus* (L.) Lam. (Rhamnaceae), a key shrub species native to North Africa, is commonly found in arid and semi-arid regions. Renowned for its resilience under harsh conditions, it forms vegetation clusters that influence the surrounding environment. These clusters create microhabitats that promote biodiversity, reduce soil erosion, and improve soil fertility. However, in agricultural fields, *Z. lotus* is often regarded as an undesirable species. This study investigated the bacterial diversity and community composition along spatial gradients around *Z. lotus* patches in barley-planted and non-planted fields. Using *16S rRNA* gene sequencing, 84 soil samples were analyzed from distances of 0, 3, and 6 m from *Z. lotus* patches. MiSeq sequencing generated 143,424 reads, representing 505 bacterial ASVs across 22 phyla. Alpha-diversity was highest at intermediate distances (3 m), while beta-diversity analyses revealed significant differences in community composition across distances (*p* = 0.035). *Pseudomonadota* dominated close to the shrub (44% at 0 m) but decreased at greater distances, whereas *Bacillota* and *Actinobacteriota* displayed distinct spatial patterns. A core microbiome comprising 44 ASVs (8.7%) was shared across all distances, with the greatest number of unique ASVs identified at 3 m. Random forest analysis highlighted *Skermanella* and *Rubrobacter* as key discriminatory taxa. These findings emphasize the spatial structuring of bacterial communities around *Z. lotus* patches, demonstrating the shrub’s substantial influence on bacterial dynamics in arid ecosystems.

## 1. Introduction

The wild jujube shrub (*Ziziphus lotus* (L.) Lam.), widely distributed in arid and semi-arid regions, plays a pivotal role in shaping Moroccan ecosystems [1]. It is well known for its resilience; the shrub can withstand various abiotic and biotic stresses [2], including the infection by clover dodder *Cuscuta epithymum* (L.) L. [3], enabling it to form vegetation clusters that impact the surrounding environment. As a nurse plant, the wild jujube shrub contributes to ecological stability in arid regions. Its dense canopy provides shade, reduces surface temperatures, and retains moisture in the soil [4,5]. These microhabitats facilitate the establishment of other plant species and offer refuge to insects, birds, and small mammals [6]. Additionally, its deep root system stabilizes soil, mitigates erosion, and enhances nutrient cycling, enriching the surrounding vegetation [7]. The shrub also contributes to the formation of fertile islands by improving soil nutrient content, thereby supporting the growth of neighboring plant species [8]. Given its ecological importance and capacity to thrive in harsh environments, *Z. lotus* has the potential as a keystone species for the conservation and restoration of degraded arid and semi-arid ecosystems in Morocco and other regions facing similar environmental challenges [9].

Despite the growing number of research investigations on *Z. lotus,* the majority of studies have focused on its phytochemical composition, nutritional value, and pharmacological properties [4,10,11]. These investigations have highlighted the plant’s rich bioactive compounds and its potential applications in traditional medicine and as a functional food. However, there is a significant knowledge gap regarding the microbial communities associated with *Z. lotus*, particularly those explored through advanced molecular techniques such as 16S rRNA metabarcoding or OMICS technologies. Existing studies on *Z. lotus* microbiota are limited to the isolation and characterization of specific bacterial strains, such as phosphate-solubilizing bacteria from the plant’s rhizosphere [12] or plant growth-promoting rhizobacteria [13]. Fahsi et al. [12] successfully isolated phosphate-solubilizing bacteria from the rhizosphere of *Z. lotus* in Morocco, identifying strains belonging to the genera *Pseudomonas*, *Bacillus*, and *Enterobacter*. These strains demonstrated the ability to enhance plant growth and seed germination in wheat, as well as improve zinc absorption, highlighting their potential as biofertilizers [13]. Such bacteria play a vital role in improving soil fertility and supporting plant development by increasing the bioavailability of phosphorus, a crucial nutrient in arid and semi-arid soils. Despite these promising findings, there remains a lack of comprehensive studies that fully explore the microbial diversity associated with *Z. lotus*, particularly encompassing both culturable and non-culturable microorganisms. However, a recent study by Radouane et al. [3] investigated the endophytic bacterial communities within the shoots of *Z. lotus* using *16S rRNA* gene sequencing targeting the V5-V6 regions. This study revealed that the endophytic bacteriome was predominantly composed of phyla *Pseudomonadota*, *Bacillota*, and *Actinobacteriota*, with genera such as *Cutibacterium*, *Staphylococcus*, and *Acinetobacter* being the most abundant. Interestingly, the study also uncovered a shared endophytic bacterial composition between *Z. lotus* and its parasitic plant *Cuscuta epithymum*, suggesting a potential transmission of endophytes between the two species via haustorial connections. This discovery provided new insights into plant–microbe interactions and underscores the need for further research to understand the ecological and functional roles of these microbial communities in arid and semi-arid ecosystems [3].

In contrast to the ecological benefits of the wild jujube shrub, it poses several challenges in agricultural settings [14]. Its aggressive growth and extensive root systems compete with crops for water and nutrients, particularly in water-scarce regions. This competition can significantly reduce crop yields and increase the costs associated with land management. Moreover, the shrub’s thorny branches can hinder farming operations and damage the machinery, making cultivation and harvesting more labor-intensive. To reconcile the ecological benefits of wild jujube shrubs with their challenges in agriculture, management strategies must be context-specific. In natural ecosystems, conserving these shrubs should take precedence to support biodiversity and maintain ecosystem functions. In agricultural settings, however, integrated management practices, such as selective removal, controlled grazing, and the application of herbicides, can be implemented to curb their spread. Furthermore, educating farmers about the shrub’s impacts and offering alternative solutions can promote sustainable land use practices.

The present study aims to determine whether wild jujube shrubs in agricultural fields can act as reservoirs of bacterial communities that play important ecological roles in arid ecosystems. The specific objectives were to (1) characterize bacterial diversity and community composition associated with *Z. lotus* across spatial gradients using 16S rRNA metabarcoding; (2) assess variations in these communities based on proximity to the shrub in barley-planted and non-planted fields; and (3) identify key bacterial taxa and their potential ecological functions.

We hypothesize that (1) bacterial richness and community structure exhibit spatial distribution patterns around wild jujube shrubs in both cultivated and non-cultivated fields and (2) bacterial diversity will vary with proximity to the shrub, potentially being elevated in soils influenced by the shrub compared to those farther away. To test these hypotheses, we conducted a sampling campaign during the crop harvest period, which coincided with the flowering stage of the wild jujube shrubs. Sampling was carried out in two adjacent fields, each approximately one hectare in size: one planted with barley and the other left uncultivated. Both fields contained spontaneous wild jujube shrubs. Soil samples were collected from within each wild jujube cluster, as well as at distances of 3 m and 6 m from the shrubs. Amplicon sequencing targeting the V5–V6 regions of the bacterial *16S rRNA* gene was performed to analyze the bacterial communities.

## 2. Materials and Methods

### 2.1. Description of Sampling Site and Sample Collection

The sampling site was located in Rhamna Province (GPS coordinates: 31°59′44″ N, 8°01′02″ W), approximately 100 m from Highway A3. The site consisted of two adjacent fields, each covering approximately 1 hectare. One field was planted with barley, while the other was left uncultivated. The planted field was occasionally irrigated during the growing phase of barley (Appendix A).

The Rhamna region’s climate represents a transition between the Mediterranean climate to the north and the more arid conditions to the south. The climate is classified as semi-arid to arid, characterized by long, hot, and dry summers. Daytime temperatures during peak summer months (July and August) can exceed 40 °C, with minimal rainfall often resulting in drought conditions. Winters are relatively mild and short, with average temperatures ranging from 10 °C to 15 °C, and nighttime temperatures occasionally dropping below 5 °C.

The region experiences limited annual rainfall, averaging between 200 mm and 400 mm. Most precipitation occurs between November and March, often in the form of occasional heavy showers. Summers are almost entirely dry, contributing to the arid nature of the landscape.

*Ziziphus lotus* is a dominant perennial shrub species that forms characteristic vegetation patches, accumulating wind-borne sediments. On 16 May 2023, sampling was conducted across two adjacent fields situated on a flat plateau to minimize topographic effects. A spatially explicit sampling design was employed to assess the impact of *Z. lotus* patches on soil bacterial communities (Figure 1). In the barley-planted field, five patches were sampled, while two patches were sampled in the non-planted field. For each patch, twelve soil samples were collected at a depth of 20 cm. Weighing approximately 1 Kg each, the samples were collected following the scheme illustrated in Figure 1. The sampling followed a distance-based design: four samples within the *Z. lotus* patch (0 m), four samples at a 3 m distance from the patch edge, and four samples at a 6 m distance. These distances were determined based on the minimal separation between two patches, which was 12 m. In total, 84 soil samples were collected (60 from the barley-planted field and 24 from the non-planted field). The samples were placed in Ziploc plastic bags (30 × 20 cm), kept on ice, and transported to the laboratory for further analysis.

### 2.2. Soil Physicochemical Analysis

The physicochemical characterization of the soil was conducted on composite samples representing each sampling cluster (each sample comprised a composite of four subsamples). For each *Z. lotus* patch, soil samples collected at varying distances (0 m, 3 m, and 6 m) were homogenized to create representative composite samples. These composite samples were analyzed at the AITTC laboratory at Mohammed VI Polytechnic University (Benguerir, Morocco). The analysis included measurements of 20 soil parameters, such as granulometry (sand, silt, and clay), pH, electrical conductivity (EC), nitrogen (NO^3−^, NH^4+^, and total N), P_2_O_5_, K_2_O, CaO, CaCO_3_, Fe, Zn, Cu, Mn, Na_2_O, MgO, organic matter (OM), and cation exchange capacity (CEC).

### 2.3. DNA Extraction and Polymerase Chain Reaction (PCR) Quantification

Total genomic DNA was extracted from 250 mg of soil using the Soil Pro Kit (Qiagen, Global Diagnostic Distribution, Témara, Morocco) according to the manufacturer’s protocol. Prior to extraction, the samples were homogenized using a TissueLyser II with 2 mm Tungsten beads (Qiagen) at 24 Hz for 15 min. DNA quality and quantity were assessed using agarose gel electrophoresis and spectrophotometric measurements on a BioSpectrophotometer (Eppendorf, Hamburg, Germany).

The V5-V6 hypervariable regions of the bacterial *16S rRNA* gene were used with a primer pair that included custom sequence (CS) adapters at the 5′ end: CS1-719F/ACACTGACGACATGGTTCTACA-AACMGGATTAGATACCCKG and CS2-1115R TACGGTAGCAGAGACTTGGTCT-AGGGTTGCGCTCGTTG [15]. Each 25 μL PCR reaction contained 1X Platinum Direct PCR Universal Master Mix (Thermo Fisher, Rabat, Morocco), 0.2 μM of each primer, and approximately 10 ng of template DNA. Amplification was performed in duplicate using a Mastercycler X50s (Eppendorf, Hamburg, Germany) with the following conditions: initial denaturation at 94 °C for 3 min, followed by 35 cycles of 94 °C for 30 s, 55 °C for 30 s, and 72 °C for 1 min, with a final extension at 72 °C for 5 min. Negative controls (sterile Milli-Q water) and positive controls were included in each PCR run.

### 2.4. Library Preparation and Sequencing

The bacterial *16S rRNA* gene amplicon library preparation was performed as described in Morales-Briones et al. [16]. Briefly, the PCR products were purified using Agencourt AMPure XP beads (Beckman Coulter, Brea, CA, USA). Purified PCR products were then resuspended in 10 mM Tris (pH 8.5). A second PCR was performed to attach Illumina sequencing adapters and index tags. PCRs for indexing contained 5 µL of purified PCR product, 2.5 µL of Fluidigm Access Array Barcode 384, and 1X KAPA HiFi HotStart ReadyMix (Roche Sequencing Solutions, Santa Clara, CA, USA). The PCR volume was 50 µL per reaction and was run under the following conditions: an initial denaturation at 95 °C for 3 min, followed by 8 cycles of denaturation at 95 °C for 30 s, annealing at 55 °C for 30 s, extension at 72 °C for 30 s, and a final extension at 72 °C for 5 min. The indexed amplicons were subsequently purified using Agencourt ampure XP beads and quantified using a Qubit assay and the DNA HS kit (ThermoFisher, Témara, Morocco). Library quantification, normalization, and pooling were performed following Illumina’s instructions. The bacterial *16S rRNA* gene libraries were sequenced on an Illumina MiSeq sequencing instrument (Illumina, Paris, France) using a MiSeq reagent V3 kit (300 cycles of paired-end sequencing).

### 2.5. Bioinformatic Analysis

Raw sequence data was processed using the DADA2 pipeline in R v4.3.3. [17]. The initial quality assessment eliminated reads with quality scores below 30. After primer and adapter removal, forward reads were filtered and denoised using DADA2’s error model. Forward reads longer than 10 bp were clustered into amplicon sequence variants (ASVs). Taxonomic classification of ASVs was performed using the latest SILVA reference database [18].

### 2.6. Statistical Analysis

To normalize the depth of sequencing, the ASVs’ abundances in each sample were transformed into compositional data using the transform command of the phyloseq package with the “compositional” parameter.

Bacterial alpha-diversity was calculated through the Shannon and Simpson indices at the ASV level using the phyloseq package [19]. Beta-diversity was assessed by computing the Bray–Curtis dissimilarity across different microbial taxa, and it was tested through PERMANOVA using the adonis command of the vegan package [20]. The command betadisper from the vegan package was also used to test the difference in dispersion between different conditions.

Random forest models were created using the randomForest package of R [21] and 100 repeated trees. The best models created with this package were then evaluated in terms of the strength of their predictive power with respect to discriminating between conditions based on communities; the best predictor taxa were listed. Significant associations of taxa with respect to the conditions were calculated using the indicspecies package [22]. The differential analysis was conducted using the STAMP package [23]. Pearson correlation analyses were performed between the abundances of the top 20 ASVs and soil variables (e.g., soil pH, nutrients) to generate the heatmap.

## 3. Results

### 3.1. Soil Physicochemical Characteristics

Appendix A presents the results of the physicochemical analysis of various samples. Multivariate regression–ANOVA (Appendix A) demonstrated that no measured edaphic variable showed a significant effect on Shannon diversity (all *p* > 0.05), suggesting that, within the range of conditions studied, soil chemistry did not strongly influence overall species richness. In contrast, the Simpson evenness index responded detectably to the silt fraction (*p* = 0.04), implying that finer textures favor a more balanced distribution of taxa even when species counts remain constant. All other factors, including pH, salinity, and major nutrients, fell well above the 0.05 threshold, underscoring the limited collective influence of soil chemistry on alpha-diversity when texture is accounted for simultaneously.

Univariate regression–ANOVA (Appendix A) refined this picture. Organic matter content emerged as the sole driver of Shannon richness (F = 4.95, *p* = 0.03), highlighting the primacy of carbon inputs for maintaining a larger species pool. Simpson evenness proved more sensitive, increasing with higher silt and sand percentages, near-neutral pH, elevated organic matter, and several macro-nutrients (P_2_O_5_, NO_3_^−^, and total N) as well as Mn (all *p* ≤ 0.05). Together, these findings reveal a dichotomy: richness is governed chiefly by labile organic carbon, whereas the relative abundances of taxa are reshaped by both particle size distribution and nutrient availability. Hence, management practices that raise organic matter and maintain balanced nutrient profiles—particularly in finer-textured soils—are likely to promote not only a diverse but also an ecologically even microbial community.

Table 1 reveals that environmental factors such as total P and total K significantly affect community structure at a distance of 0 m, with their influence diminishing at greater distances (3 m and 6 m), reflecting spatial variability.

### 3.2. Bacterial Diversity and Richness Respond to Spatial Distribution

Sequencing analysis yielded a total of 143,424 reads, averaging 1707 reads per sample across all datasets. These reads were classified into 505 bacterial Amplicon Sequence Variants (ASVs), which were assigned to 146 genera, 111 families, 89 orders, 46 classes, and 22 phyla.

Analysis of bacterial communities revealed distinct spatial patterns in taxonomic composition across the sampling transect. The predominant phylum, *Pseudomonadota*, exhibited a decreasing abundance gradient from 44% at 0 m to 27% at 3 m and 6 m. *Bacillota* and *Actinobacteriota* were the next most prevalent phyla, comprising 27% and 21% of sequences, respectively. *Bacillota* exhibited relatively stable abundances across all distances. Among these major phyla, *Actinobacteriota* displayed the most notable spatial variation, with higher relative abundances detected at the 0 m sampling point (21%) compared to the 3 m and 6 m locations (10–15%). The communities also contained lower proportions of other phyla, including *Bacteroidota* and candidate phyla, which collectively represented less than 8% of total sequences (Figure 2).

Alpha-diversity analyses revealed significant differences in bacterial community structure across sampling distances. Shannon diversity indices showed significant but moderate variation between distances (PERMANOVA, F = 4.235, *p* = 0.018) (Table 2). Pairwise comparisons demonstrated that samples at 3 m had significantly higher Shannon diversity compared to those at 0 m (*p* = 0.026) and 6 m (*p* = 0.004). However, no significant differences were observed between communities at 0 m and 6 m (*p* = 0.205). These results suggest that bacterial diversity peaks at intermediate distances in this sampling design, with similar lower diversity levels at both 0 m and 6 m sampling points. This pattern was consistent across both Shannon and Simpson diversity indices, indicating a robust mid-distance peak in community diversity. Statistical analyses clearly demonstrate that distance has a significant effect on bacterial community structure, with optimal diversity occurring at intermediate spatial scales. Meanwhile, no significance was observed when comparing the planted and non-planted fields in both indices (Figure 3 and Table 3).

### 3.3. Bacterial Communities Are Predominantly Shaped by Distance

Beta-diversity analyses indicate that the two PCoA axes capture only a small portion of the Bray–Curtis variance (Axis 1 = 7.2%; Axis 2 = 6.6%) (Figure 4). Although the 3 m samples show a slight tendency to cluster toward the upper-left quadrant, there is no pronounced separation among distances, nor between planted and non-planted fields. This visual pattern suggests that distance from the tree row and planting status exert, at most, a modest influence on overall community composition relative to the background heterogeneity of the site.

### 3.4. Common Core Bacterium Exists Across the Spatial Distribution

The analysis of shared and unique bacterial taxa revealed distinct spatial distribution patterns across the three sampling distances. Analysis of the Venn diagrams (Figure 5) confirmed that each distance harbors a distinct subset of the bacterial community, while a sizeable core persists across the transect. At the genus level, 3 m contained the greatest number of unique genera, 37 taxa (18.0%), followed by 0 m with 34 taxa (16.6%); the outermost 6 m position contributed just eight unique genera (3.9%). A further 26 genera (12.7%) were shared exclusively between 0 m and 3 m, nine genera (4.4%) between 0 m and 6 m, and 18 genera (8.8%) between 3 m and 6 m. Notably, 73 genera (35.6%) formed a common core present at all three distances.

A comparable pattern appears at the ASV level. Here, 3 m again displayed the highest number of unique sequence variants, 637 ASVs (35.5%), while 0 m and 6 m harbored 467 (26.0%) and 304 (16.9%) unique ASVs, respectively. Pairwise overlaps comprised 112 ASVs (6.2%) shared between 0 m and 3 m, 46 ASVs (2.6%) between 0 m and 6 m, and 80 ASVs (4.5%) between 3 m and 6 m. The core microbiome across the gradient encompassed 149 ASVs (8.3%), underscoring that, despite distance-specific signatures, a substantial fraction of the community remains ubiquitous along the 6 m transect.

Random forest analysis of bacterial taxa revealed distinct patterns across the sampling distances, identifying the top 20 discriminatory ASVs. *Skermanella* (ASV_3) exhibited the largest mean decrease in accuracy (13.37), followed by *Rubrobacter* species ASV_15 and ASV_5 (6.97 and 5.77, respectively). Regression analysis identified 18 ASVs significantly associated with sampling distance, particularly within the phyla *Actinobacteriota*, Proteobacteria, and Cyanobacteria. At 0 m, several ASVs of *Actinobacteriota* (e.g., ASV_16, ASV_4, ASV_14) showed positive linear coefficients. At 3 m, ASV_3 of Proteobacteria demonstrated strong positive associations. At 6 m, key ASVs included ASV_471 (*Bacillota*) and two cyanobacteria representatives (ASV_193, ASV_485).

These results show that each sampling distance harbors a distinct bacterial community, with specific phyla showing spatial preferences. The results align with beta-diversity analyses, providing taxonomic resolution to the observed community differences across spatial scales (Appendix A, Figure 6).

In Figure 7, the restructuring of bacterial community in the planted and non-planted barley field by the mean relative abundances shows that *Solirubrobacter*, *Bacillus*, *Nocardioides*, *Pseudomonas*, and *Streptomyces* dominate the planted plots, whereas *Paeniglutamicibacter*, *Adhaeribacter*, *Domibacillus*, and *Pontibacter* are more prevalent in the non-planted field. These trends are confirmed by the 95% confidence intervals for the mean–proportion differences. Genera enriched in the planted field fall entirely on the negative side of the axis, while those enriched in the non-planted field fall on the positive side. All displayed genera remained significant after multiple-test correction (*p* < 0.05), indicating that the observed shifts are robust. Collectively, these data demonstrate that barley cultivation drives a substantial and selective enrichment of rhizosphere-associated taxa, highlighting the crop’s capacity to reshape soil microbiota toward communities often linked to plant growth promotion and nutrient cycling.

### 3.5. Soil Physicochemical Factors Drive Soil Bacterial ASVs

The analysis revealed distinct patterns of correlation between specific ASVs and key soil physicochemical properties (Figure 8). Among these, ASV_300 demonstrated a strong positive correlation with salinity (EC), and a negative correlation with total P (P_2_O_5_) and total K (K_2_O), indicating its preference for saline environments but sensitivity to nutrient-rich conditions. In addition, ASV_9 and ASV_197 showed negative correlations with total K (K_2_O) and total P (P_2_O_5_), suggesting decreases in their abundance in potassium- and phosphorus-enriched soils. ASV_357 and ASV_396 exhibited negative associations with total N, highlighting their lower abundance in nitrogen-rich soils, while ASV_86 was found to be positively associated with salinity (EC), indicating its preference for high salt concentrations, like ASV_300. On the other hand, ASV_217 showed a negative correlation with total N and total P, reflecting its sensitivity to soils rich in these essential nutrients. These results underscore that salinity, total P, total K, and total N play pivotal roles in shaping the distribution of specific bacterial taxa, with ASV_300, ASV_9, ASV_197, ASV_357, ASV_396, ASV_86, and ASV_217 displaying distinct responses to these soil characteristics.

## 4. Discussion

We investigated the spatial distribution of bacterial communities around wild jujube shrub patches and evaluated their ecological roles in arid environments. Our findings revealed a complex pattern of bacterial diversity, with significant variation at certain distances from the shrub patch while remaining consistent at others. This suggests that wild jujube shrubs exert differential influences on bacterial communities beyond their immediate vicinity. These results support our first hypothesis that bacterial richness and community structure exhibit distinct spatial distribution patterns around wild jujube shrubs in both cultivated and non-cultivated fields. Furthermore, we assessed the impact of spatial distribution on bacterial assemblages and the intricate interactions between these communities and their abiotic environment.

Our results reveal a clear shift in community composition with distance, characterized by a decline in fast-growing copiotrophs (e.g., *Pseudomonadota*) near the shrub and a stable presence of oligotroph-associated groups (e.g., *Bacillota*) across all distances. This decline may be attributed to differences in nutrient content between the rhizosphere of wild jujube shrubs and the surrounding soil at greater distances [24,25,26]. The varying abundance patterns of *Actinobacteriota* suggest that they may be highly sensitive to the environmental gradients and nutrient availability [27]. In contrast, *Bacillota* maintained a relatively stable relative abundance across all distances, indicating ecological stability under the varied conditions. Less than 8% of the total sequences were composed of *Bacteroidota* and the candidate phyla, which appeared in low proportions. These findings align with the concept of copiotrophic and oligotrophic bacterial strategies, as described by Fierer et al. [27]. *Pseudomonadota*, which are considered copiotrophs, likely benefit from the higher nutrient availability at the 0 m site, while the consistent abundance of *Bacillota* suggests their ability to adapt to more stable environmental conditions. The decreasing abundance of *Actinobacteriota* with increasing distance further supports their dependence on specific, high-nutrient environments, which is consistent with observations from rhizosphere studies [28,29,30]. These spatial patterns reflect the "islands of fertility" model in landscape ecology, where shrubs such as *Z. lotus* concentrate resources and thereby support richer microbial communities under their canopies. Such nurse plants can induce plant–soil feedbacks that enhance their own ecosystem, for example by fostering microbes involved in nutrient cycling, which in turn benefits neighboring vegetation [31].

The results of this study reveal significant spatial variability in bacterial community diversity and structure along the sampling transect, with a marked peak in diversity at 3 m, as indicated by the Shannon and Simpson diversity indices. These findings partially support our second hypothesis, which proposed that bacterial diversity would be highest within the wild jujube shrub clusters and their immediate surroundings, decreasing with distance. The intermediate distance peak (3 m) is consistent with observations by Houlden et al. [30], who reported that microbial diversity in the rhizosphere is shaped by dynamic plant–root interactions, particularly during phases of heightened resource exchange. The similar diversity levels at 0 m and 6 m suggest that proximity to or distance from resource-rich zones may limit bacterial community complexity, aligning with findings that resource availability and competition strongly influence microbial dynamics [32,33]. The core bacteriome shared across distances, comprising 44 ASVs and 35 genera, highlights the resilience of the local arid soil bacterial community, as also observed in the jujube (*Ziziphus jujuba* Mill.) rhizosphere by Liu et al. [28], where core taxa persisted despite variations in environmental conditions. Although distance from the shrub had a statistically significant effect on bacterial community structure, the effect size was relatively modest. This suggests that, while distance influences the microbiome to some extent, a considerable portion of community variation is likely driven by other sources of environmental heterogeneity.

Beta-diversity analyses revealed a low shift in bacterial community composition, with ordination analyses (Bray–Curtis) showing distinct clustering at 3 m compared to 0 m and 6 m. Fierer et al. [29] identified resource availability as a driver of both taxonomic and phylogenetic variation in soil microbial communities. The increased uniqueness of taxa at 3 m reflects conditions that favor a more specialized community structure, supporting the copiotroph–oligotroph model where resource gradients influence microbial functional strategies. Although there was overlap in bacterial communities, the distinctiveness of the 3 m samples suggests that intermediate spatial scales may act as biodiversity hotspots due to optimal nutrient availability and competitive balance, as proposed in a study of rhizosphere microbial assemblages [34,35].

The spatial distribution of the top 20 taxa reveals different ecological roles and assembly processes affecting bacterial communities at various sampling distances. For example, *Skermanella* (ASV_3), which was identified as the strongest discriminatory feature linked to the largest decrease in accuracy at a particular distance, appears to thrive at the intermediate 3 m distance. Similarly, the dominance of *Rubrobacter* species (ASV_5, ASV_15) at 0 m and the enrichment of *Bacillota* and Cyanobacteria (ASV_471, ASV_193) at 6 m emphasize their preference for specific ecological conditions associated with each distance. These findings align with earlier reports suggesting that *Actinobacteriota* and *Pseudomonadota* phyla display spatially resolved distributions influenced by nutrient availability and oxygen gradients [28,36,37].

Regression analysis indicates that *Actinobacteriota* taxa at 0 m are positively associated with resource-rich microhabitats, whereas *Pseudomonadota* are strongly positively associated at 3 m, suggesting their ability to adapt to moderately rich and stable environments [38].

The relationship between taxon abundance and soil physicochemical properties suggests that soil characteristics play a role in shaping bacterial community composition. *Actinobacteriota* (ASV_1) showed a negative correlation with nitrate and potassium, which may reflect competitive interactions or nutrient depletion effects [29,39]. Soil salinity (EC) exhibited a strong positive correlation with *Verrucomicrobiota* (ASV_300), suggesting a possible specialization to saline environments. This is in accordance with previous studies that identified salinity and nutrient gradients as major influencers of microbial distribution and community assembly [36,40,41]. These results highlight the relationship between soil and microbial ecology, underscoring the importance of understanding environmental heterogeneity and its impact on microbial community dynamics. Consequently, this study provides valuable insights into soil management and restoration strategies that enhance microbial diversity and ensure functional stability within ecosystems.

## 5. Conclusions

This study reveals significant spatial variability in bacterial community diversity and structure around wild jujube shrub patches in arid environments. While the bacterial community exhibited a pronounced peak in diversity at 3 m, the findings challenge our second hypothesis, which suggested that bacterial diversity would be highest within the shrub clusters and decrease with distance. Instead, diversity appeared to increase at intermediate distances, reflecting the dynamic interactions between plant roots and microbial communities. This study highlights the role of resource gradients and ecological factors in shaping microbial communities, with specific taxa exhibiting spatial preferences linked to nutrient availability. These insights emphasize the importance of spatial scale in microbial ecology and offer valuable implications for soil management and restoration strategies aimed at enhancing microbial diversity and ecosystem stability.

In summary, preserving wild jujube shrub patches in arid agricultural fields can enhance soil microbial diversity, improve soil fertility, and support overall ecosystem stability. These ecological benefits are valuable for sustainable agriculture in water-limited, challenging environments. However, these benefits must be weighed against potential drawbacks, such as competition with crops and reduced yields. In many cases, the immediate costs to crop production may outweigh the benefits, suggesting that, in cultivated areas, careful management or selective removal could be necessary, even though maintaining some shrubs at field margins may provide long-term soil health advantages.

## Figures and Tables

**Figure 1 microorganisms-13-01740-f001:**
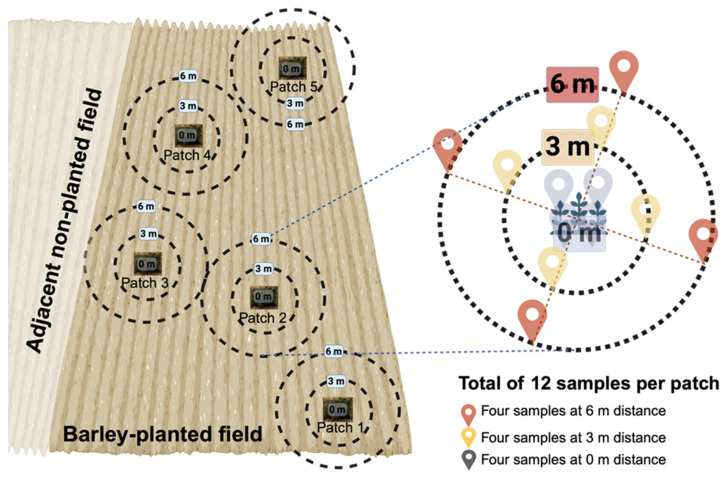
Experimental sampling in barley-planted and non-planted fields. The scheme illustrates the sampling design, with soil samples collected at three distances (0 m, 3 m, and 6 m) from Z. lotus patches. Each sampling cluster comprises a total of 12 samples.

**Figure 2 microorganisms-13-01740-f002:**
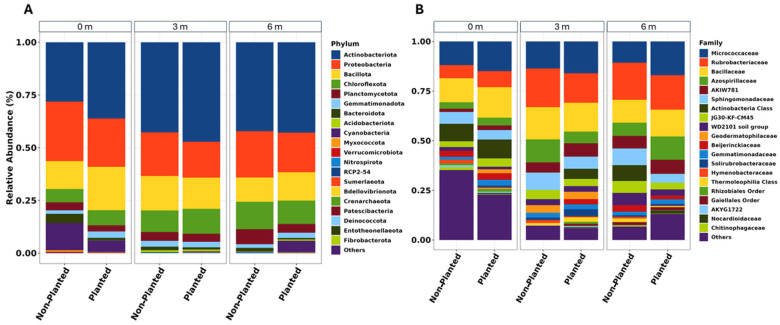
Distribution of the 20 most prevalent taxa based on the sampled distances in the planted and non-planted fields, categorized by (**A**) phylum and (**B**) family. Taxa not included in the top 20 are grouped as “Others”.

**Figure 3 microorganisms-13-01740-f003:**
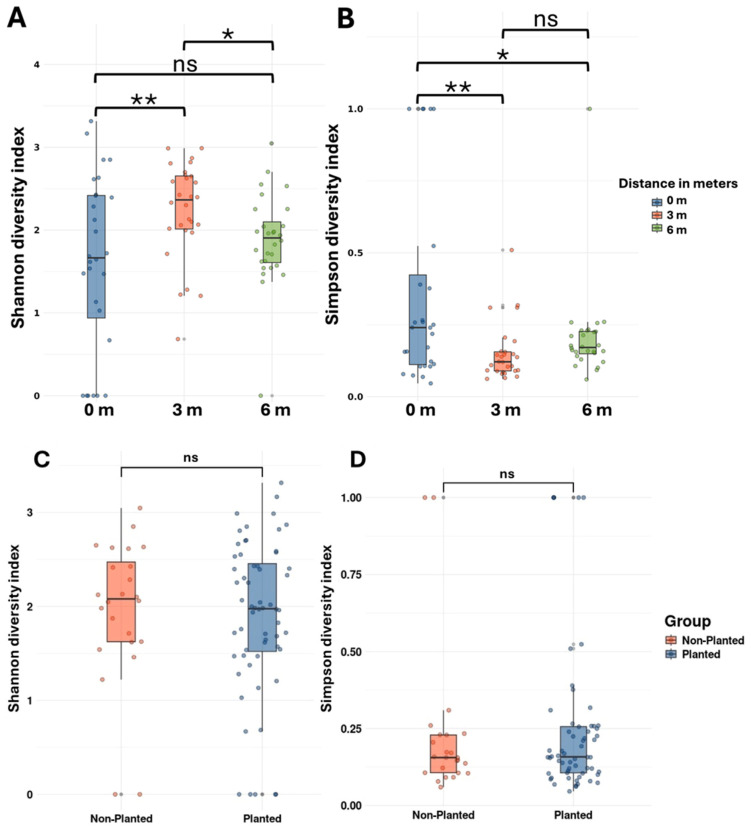
Alpha-diversity was evaluated across different distances based on Shannon (**A**) and Simpson (**B**) indices and between planted and non-planted barley fields (**C**,**D**), respectively. Asterisks (* *p* ≤ 0.05, ** *p* ≤ 0.01) indicate significant differences, while “ns” denotes non-significant results.

**Figure 4 microorganisms-13-01740-f004:**
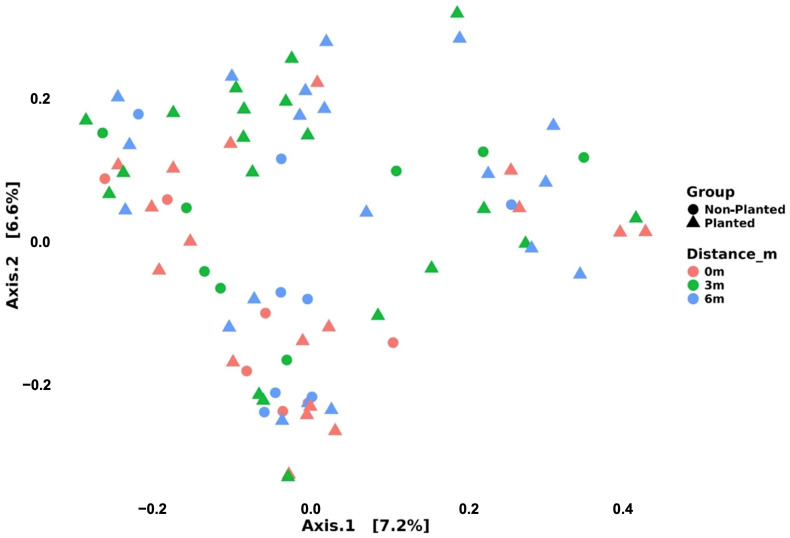
Spatial distribution patterns at the three sampling distances (0, 3, and 6 m) and the difference between planted and non-planted fields based on Principal Coordinate Analysis (PCoA) using Bray–Curtis dissimilarity.

**Figure 5 microorganisms-13-01740-f005:**
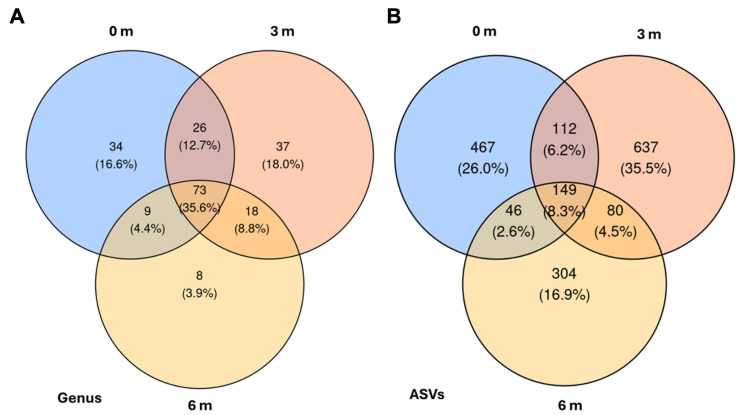
Venn diagrams illustrating the distribution of taxa across different sampling distances, categorized by (**A**) genera and (**B**) ASVs.

**Figure 6 microorganisms-13-01740-f006:**
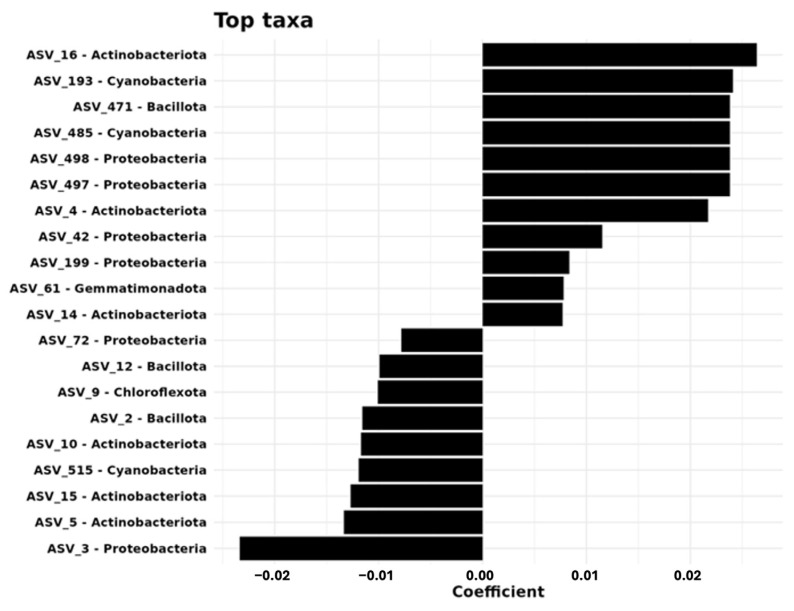
Key taxa influencing microbial community structure, identified through coefficient analysis.

**Figure 7 microorganisms-13-01740-f007:**
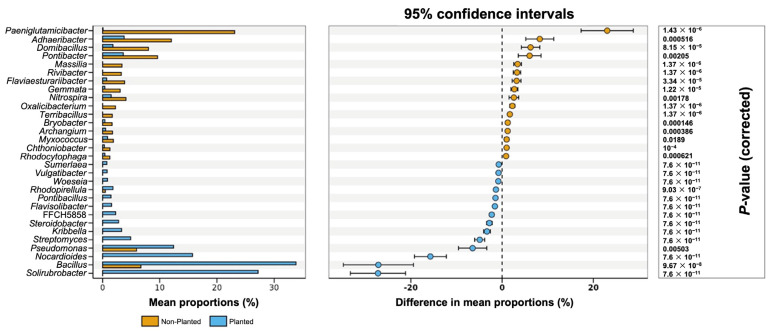
Genus-level differences in bacterial community composition between planted and non-planted fields. Bars represent the mean relative abundances of key genera for each field type, with error bars indicating 95% confidence intervals.

**Figure 8 microorganisms-13-01740-f008:**
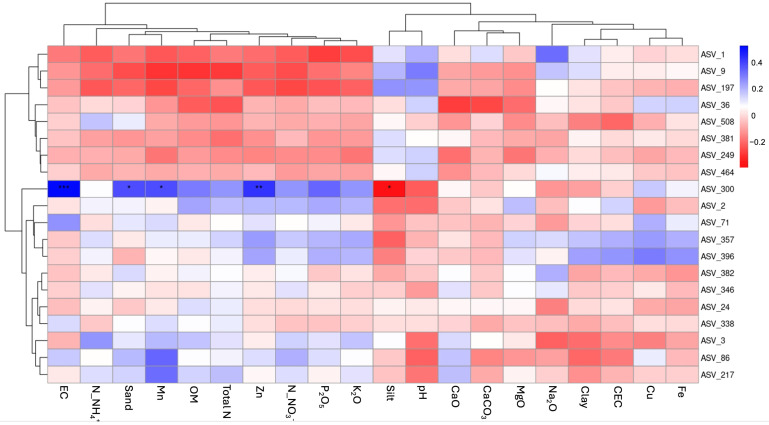
Heatmap showing correlations between soil factors and dominant bacterial ASVs. Significance levels (*p*-value) are shown, and asterisks (* *p*  ≤  0.05, ** *p*  ≤  0.01, and *** *p*  ≤  0.001) indicate the significant effect.

**Table 1 microorganisms-13-01740-t001:** Influence of selected environmental variables on community structure at different distance levels (0 m, 3 m, and 6 m). Significance levels (*p*-values) are provided, with asterisks indicating significant differences (* *p* ≤ 0.05, ** *p* ≤ 0.01).

Distance in Meters	Factor	*p*_Value	R2	Chi_Square_Percent	Significance
0 m	Total P	0.007	0.057884	5.78844	**
	Total K	0.042	0.047146	4.714648	*
	Total N	0.744	0.032035	3.203506	
3 m	Total P	0.351	0.039211	3.921062	
	Total K	0.345	0.039303	3.930348	
	Total N	0.921	0.027068	2.706801	
6 m	Total P	0.671	0.032995	3.299475	
	Total K	0.817	0.030101	3.010063	
	Total N	0.5	0.036909	3.690935	

**Table 2 microorganisms-13-01740-t002:** PERMANOVA test at the phylum level using the Aitchison method. Significance levels (*p*-values) are provided, with asterisks (* *p* ≤ 0.05) indicating significant results.

Df	Sum Sq	Mean Sq	F Value	Pr (>F)
2	0.038003	0.019001	4.234772	0.017815 *
81	0.36345	0.004487		

**Table 3 microorganisms-13-01740-t003:** Factors significantly influencing alpha-diversity, assessed by the Shannon index, across different sampling distances. Significance levels (*p*-values) are presented, with asterisks (* *p* ≤ 0.05, ** *p* ≤ 0.01) indicating significant differences, while “ns” denotes non-significant results.

Index	Comparison	*p*_Value	Adjusted_*p*_Value	Significance
Shannon	0 m vs. 3 m	0.025625	0.025625	*
Shannon	0 m vs. 6 m	0.205278	0.205278	ns
Shannon	3 m vs. 6 m	0.00425	0.00425	**

## Data Availability

The raw reads for the five clusters from the barley-planted fields and the two clusters from the non-planted field samples were submitted to NCBI (https://www.ncbi.nlm.nih.gov/) under BioProject ID: PRJNA1203891.

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
