# Peer review of "Influence of Ziziphus lotus (Rhamnaceae) Plants on the Spatial Distribution of Soil Bacterial Communities in Semi-Arid Ecosystems"

_microorganisms, 2025, doi:10.3390/microorganisms13081740_

Round 1

Reviewer 1 Report

Comments and Suggestions for Authors

This paper discusses the spatial structuring of bacterial communities around Z. lotus patches, comparing cultivated and non-cultivated areas.

Figure 3D - y-axis should be Simpson

Tables 2 and 3: Commas were used instead of points

Figure legend 7: was missing

Methods: missing methods for figure 7+8

Different fonts within the text - for example lines 50-54

Reference  #17 - not related

Reference that was not mentioned: https://doi.org/10.1016/j.jaridenv.2021.104614

"Preserving wild jujube shrub patches in cultivated fields in arid and semi-arid envi- 472
ronments is essential, as they contribute to enhancing soil microbial diversity, improving 473
soil fertility, and supporting ecosystem stability, all of which are vital for sustainable ag- 474
ricultural practices in these challenging environments. " - what about the "costs" described in the introduction associated with these wild jujube shrub in an agricultural setting? Do these benefits outweigh the costs? 

Author Response

Comment 1: [Figure 3D - y-axis should be Simpson.]

Response 1: [changed.]

Comment 2: [Tables 2 and 3: Commas were used instead of points.]         

Response 2: [we replaced comma by points.]

Comment 3: Figure legend 7: was missing

Response 3: [we added figure 7 captation.]

Comment 4: [Methods: missing methods for figure 7+8.]

Response 4: [We included the methods used for both figures in the Materials and Methods section, along with the appropriate reference.]

Comment 5: [Different fonts within the text - for example lines 50-54.]

Response 5: [done.]

Comment 6: [Reference  #17 - not related

Reference that was not mentioned: https://doi.org/10.1016/j.jaridenv.2021.104614.]

 Response 6: [we replaced the reference.]

Comment 7: ["Preserving wild jujube shrub patches in cultivated fields in arid and semi-arid environments is essential, as they contribute to enhancing soil microbial diversity, improving soil fertility, and supporting ecosystem stability, all of which are vital for sustainable agricultural practices in these challenging environments. " - what about the "costs" described in the introduction associated with these wild jujube shrub in an agricultural setting? Do these benefits outweigh the costs?]

Response 7: [We fully agree with the reviewer’s comment. To address this concern, we have explicitly weighed the benefits versus the costs of wild jujube shrubs in farmlands. As noted in the Introduction, these shrubs provide ecological benefits, such as enhancing biodiversity, soil fertility, and microclimate, but also pose agricultural challenges, including competition with crops.]

Reviewer 2 Report

Comments and Suggestions for Authors

Dear Authors,

This study addresses a relevant and timely topic by examining plant–microbiome interactions in resource-limited environments using a well-structured spatial sampling approach and 16S rRNA metabarcoding. This manuscript presents a scientifically sound investigation of ecological relevance. However, the following revisions are required to enhance the clarity, interpretive depth, and structural quality of the manuscript.

Line 44: Revise to: “keystone species in the conservation…”

Lines 51–74: The background section effectively frames the research context, but is somewhat verbose and partially repetitive. Please condense this portion and better emphasize the gap your study fills, particularly regarding Ziziphus lotus as a keystone species in microbiome-driven ecosystem stability.

Lines 95–98: The second hypothesis appears to contradict your observed results (no peak at 3 m). Please rephrase this to reflect either an exploratory nature or to clarify that multiple outcomes were possible a priori.

Line 106: Correct the heading “Materials and Methodes” → “Materials and Methods”

Line 159: Provide a citation/source for the primer pair and adapters beyond Radouane et al., as CS tags are nonstandard.

Lines 211–230: Avoid overinterpreting non-significant statistical results. Statements about the “limited influence of soil chemistry” should be cautiously framed, especially when results are marginal or non-significant (e.g., p > 0.05).

Lines 291–296: Your statement that distance has a “modest influence” contradicts the statistical significance shown earlier. Please reconcile these statements for consistency.

Lines 379–423: The Discussion repeats many earlier results without a deeper ecological synthesis. Consider reducing redundancy and introducing more conceptual interpretations (e.g., landscape ecology principles, plant–soil feedbacks, or core–satellite models).

Lines 430–447: Avoid definitive functional claims (e.g., “nutrient-driven adaptation”) unless supported by metagenomic or biochemical data. Use cautious phrasing such as “suggests functional specialization.”

Line 471: Revise phrasing in the final sentence of the conclusion for conciseness.

Author Response

Comment 1: [Line 44: Revise to: “keystone species in the conservation…”]

Response 1: [Corrected.]

Comment 2: [Lines 51–74: The background section effectively frames the research context, but is somewhat verbose and partially repetitive. Please condense this portion and better emphasize the gap your study fills, particularly regarding Ziziphus lotus as a keystone species in microbiome-driven ecosystem stability.]

Response 2: [We thank the reviewer for his comment; correction was made.]

Comment 3: [Lines 95–98: The second hypothesis appears to contradict your observed results (no peak at 3 m). Please rephrase this to reflect either an exploratory nature or to clarify that multiple outcomes were possible a priori.]

Response 3: [Corrected.]

Comment 4: [Line 106: Correct the heading “Materials and Methodes” → “Materials and Methods”]

Response 4: [Corrected.]

Comment 5: [Line 159: Provide a citation/source for the primer pair and adapters beyond Radouane et al., as CS tags are nonstandard.]

Response 5: [We agree with the reviewer’s comment; a new citation was added.]

Comment 6: [Lines 211–230: Avoid overinterpreting non-significant statistical results. Statements about the “limited influence of soil chemistry” should be cautiously framed, especially when results are marginal or non-significant (e.g., p > 0.05).]

Response 6:  [We have revised the text to address all the identified issues .]

Comment 7: [Lines 291–296: Your statement that distance has a “modest influence” contradicts the statistical significance shown earlier. Please reconcile these statements for consistency.]

Response 7:  [We revised the sentence to improve clarity and eliminate contradictions.]

Comment 8: [Lines 379–423: The Discussion repeats many earlier results without a deeper ecological synthesis. Consider reducing redundancy and introducing more conceptual interpretations (e.g., landscape ecology principles, plant–soil feedbacks, or core–satellite models).]

Response 8: [We thank the reviewer for their valuable suggestions, which have been carefully incorporated into this revision.]

Comment 9: [Lines 430–447: Avoid definitive functional claims (e.g., “nutrient-driven adaptation”) unless supported by metagenomic or biochemical data. Use cautious phrasing such as “suggests functional specialization.”]

Response 9: [We rephrased those sentences and toned down the claim as suggested by the reviewer.]

Comment 10: [Line 471: Revise phrasing in the final sentence of the conclusion for conciseness.]

Response 10: We rephrased that sentence.]